# Sex-Based Differences in Clinical Profile and Complications among Individuals with Type 2 Diabetes Seen at a Private Tertiary Diabetes Care Centre in India

**DOI:** 10.3390/healthcare11111634

**Published:** 2023-06-02

**Authors:** Rajendra Pradeepa, Lal Shreya, Ranjit Mohan Anjana, Saravanan Jebarani, Ulagamathesan Venkatesan, Nithyanantham Kamal Raj, Onkar C. Swami, Viswanathan Mohan

**Affiliations:** 1Madras Diabetes Research Foundation, ICMR Centre for Advanced Research on Diabetes, Chennai 600086, India; 2Dr. Mohan’s Diabetes Specialities Centre, Chennai 600086, India; 3Emcure Pharmaceuticals Ltd., Pune 411057, India

**Keywords:** gender, women, type 2 diabetes, clinical profile, complications, India

## Abstract

This study aimed to compare the clinical and biochemical profiles as well as the complications in males and females with type 2 diabetes (T2DM) presenting to a private tertiary diabetes care centre in India. This is a retrospective study, conducted between 1 January 2017 and 31 December 2019, and included 72,980 individuals with T2DM, aged ≥ 18 years (age and sex-matched—males—36,490; females—36,490). Anthropometric measurements, blood pressure, fasting plasma glucose (FPG), post-prandial plasma glucose (PPPG), glycated haemoglobin (HbA1c), lipids, urea, and creatinine were measured. Retinopathy was screened using retinal photography, neuropathy using biothesiometry, nephropathy measuring urinary albumin excretion, peripheral vascular disease (PVD) using Doppler, and coronary artery disease (CAD) based on the history of myocardial infarction and/or drug treatment for CAD and/or electrocardiographic changes. Obesity (73.6% vs. 59.0%) rates were significantly higher in females compared to males. FPG, PPPG, and HbA1c were higher among younger age groups among both sexes, with males having higher values compared to females. However, after the age of 44 years, control of diabetes was worse among females. In addition, only 18.8% of the females achieved glycemic control (HbA1c < 7%) compared to 19.9% in males (*p* < 0.001). Males had higher prevalence of neuropathy (42.9% vs. 36.9%), retinopathy (36.0% vs. 26.3%), and nephropathy (25.0% vs. 23.3%) compared to females. Males had 1.8- and 1.6-times higher risk of developing CAD and retinopathy compared to females. Hypothyroidism (12.5% vs. 3.5%) and cancers (1.3% vs. 0.6%) were significantly higher in females compared to males. In this large sample of T2DM seen at a chain of private tertiary diabetes centres, females had higher prevalence of metabolic risk factors and poorer diabetes control compared to males, emphasizing the need for better control of diabetes in females. However, males had higher prevalence of neuropathy, retinopathy, nephropathy, and CAD compared to females.

## 1. Introduction

Diabetes is a rapidly emerging public health problem that has reached alarming proportions in low- and middle-income countries like India [1]. The global burden of diabetes currently stands at 537 million people, of which India alone accounts for 74 million [2,3].

A slightly higher prevalence of diabetes has been noted in males compared to females (10.8% vs. 10.2%) [3]. A similar trend is observed in the Indian population in the National Family Health Survey (NHFS) with a higher proportion of males than females having random blood glucose > 140 mg/dL (30.1% vs. 25.9%) [4]. The Indian Council of Medical Research—India Diabetes (ICMR-INDIAB) study also reported that the prevalence of diabetes was significantly higher in males than in females between the ages of 35 and 65 years; however, beyond this age group, the prevalence was slightly higher in females, probably reflecting survivor bias [5].

Sex differences could be due to biological differences between males and females that are produced by variations in chromosomes, autosomal gene expression, sex hormones, and their impact on various body systems [6]. Gender is a multifaceted construct, and various gender-related traits may have varying effects on health-related behaviour and other factors, such as susceptibility to stress [7]. There are differences in the behaviour of females and males, exposure to certain environmental factors, and differences in lifestyle factors like physical activity, diet, or stress that leads to gender disparities in the risk of diabetes [6]. Studies of gender differences are also essential for planning awareness programs and understanding the burden of diabetes in females [6,8].

Current evidence suggests there are clinically significant sex variations in type 2 diabetes mellitus (T2DM) rates in youth and middle age [9,10,11,12,13,14]. There are also gender disparities in the risk of cancer, dementia, and kidney diseases in those with T2DM [15,16,17]. A recent national study has revealed that compared to men, women did not achieve lipid treatment targets, suggesting gender disparities in affordability and accessibility of treatment [18]. Hence, we hypothesize that women are likely to have suboptimal diabetes management and thereby greater propensity to develop complications. Thus, the objective of this study was to analyse the sex-based differences in demography, lifestyle, glycemic and biochemical parameters, and complications profiles in Asian Indian adults with T2DM seen at a private tertiary diabetes care centre with branches across India.

## 2. Materials and Methods

This was a retrospective study conducted on individuals with T2DM, who were registered at a tertiary diabetes care centre in the private sector with branches across India. The centre has clinical records for over 550,000 diabetes patients. For this study, data of individuals with T2DM seen between 1 January 2017 and 31 December 2019 were analysed. The inclusion criteria for this study were individuals with T2DM belonging to both sexes, aged ≥ 18 years and in whom glycemic parameters and details of complications were available for analysis. Pregnant and lactating women and individuals who had not given prior informed consent to use their data for research purposes were excluded from this study. Thus, this study included the de-identified data of 72,980 individuals with T2DM (male: 36,490; female: 36,490) who were age and gender-matched. T2DM was diagnosed by the absence of ketosis, good b-cell reserve as shown by fasting [C-peptide assay] [>0.6 pmol/mL], and stimulated C peptide > 1.0 pmol/mL and response to oral hypoglycemic agents (OHA’s) for at least 2 years [19].

On enrolment, demographic profiles and anthropometric measurements such as height, weight, and blood pressure were measured using standard methods. Height was measured in centimetres using a stadiometer and weight in kilograms using an electronic weighing scale. Weight in kg/(height in m^2^) was used to determine the body mass index (BMI). Blood pressure was measured in the right arm in the sitting posture using an electronic blood pressure apparatus. A comprehensive medical history, including current medicines and family history of diabetes, was also collected. During the first visit, biochemical parameters including fasting plasma glucose (FPG) and postprandial plasma glucose (PPPG), HbA1c, and lipid profile were measured, and diabetes complications were also were assessed as described below. At each visit, all of these parameters are archived in the diabetic electronic medical record (DEMR) [20,21].

After an overnight fast of 8 to 10 h, a fasting blood sample was taken. Participants were given a meal (carbohydrates around 60 g) and a venous blood sample was obtained after 90 min for the post-prandial glucose sample. The samples were analyzed at the clinical laboratory of the centre, which is accredited by the College of American Pathologists (CAP) and the National Accreditation Board for Testing and Calibration Laboratories (NABL). The Beckman Coulter AU 2700/480 Autoanalyzer [Beckman AU(Olympus), Ireland] was used to measure plasma glucose (glucose oxidase-peroxidase method), serum cholesterol (cholesterol oxidase phenol 4-aminoantipyrine peroxidase method), serum triglycerides (glycerol phosphate oxidase-peroxidase-amidopyrine), and high-density lipoprotein [HDL] cholesterol (direct immunoinhibition method). The low-density lipoprotein (LDL) cholesterol levels were calculated using the Friedewald algorithm. The HPLC technique was used to calculate HbA1c using a Variant machine (Bio-Rad, Hercules, CA, USA).

The complications screening included retinopathy, nephropathy, neuropathy, peripheral vascular disease [PVD], and coronary artery disease [CAD]. To identify retinopathy, seven-field stereo colour retinal photography (Zeiss FF450 Plus Digital Fundus Camera, Carl Zeiss Meditec, Inc. Dublin, Ireland,) was performed and the images were graded by an ophthalmologist using the Early Treatment Diabetic Retinopathy Study criteria [22]. The presence of at least one distinct microaneurysm was the minimum requirement for diagnosing diabetic retinopathy. Nephropathy was defined as urine albumin excretion of ≥30 µg/mg creatinine [23]. Vibratory perception thresholds (VPT) of both great toes were measured using a biothesiometer, and neuropathy was diagnosed if the mean VPT was ≥20 V [24]. PVD was diagnosed if the ankle-brachial index was ≤0.9 using Doppler investigations [25]. To assess CAD, a previous history of documented myocardial infarction and/or drug treatment for CAD (aspirin or nitrates) and/or suggestive electrocardiographic changes such as ST segment depression and/or Q-wave changes and/or T-wave changes [26].

The study was approved by the institutional Ethics Committee of the Madras Diabetes Research Foundation. All patients were requested to sign a consent form at their initial visit giving permission for their anonymised data to be used for research purposes. Only data from those patients who had provided written informed consent was included in this study.

## 3. Statistical Analysis

The data were analysed using the SAS statistical package (version 9.0; SAS Institute, Inc., Cary, NC, USA). Estimates were expressed as mean ± standard deviation or proportions. A Z test was used to compare groups for continuous variables and a chi-square test was used to compare proportions between two groups. Logistic regression analysis was used to assess the risk for complications with respect to sex, adjusting for age, HbA1c, and duration of diabetes. *p*-value < 0.05 was considered statistically significant.

## 4. Results

From the electronic records, a total of 99,192 individuals with T2DM (male: 62,691; female: 36,501) who visited the main centre or its branches from January 2017 to December 2019 and met the inclusion criteria were identified. For the final analysis, 72,980 individuals (i.e., males: 36,490; females 36,490) who were age and gender-matched were included in the study so that the two groups were comparable.

The sex-wise demographic and clinical characteristics of the study population are depicted in Table 1. Females had a higher BMI than males. Smoking and alcohol consumption were significantly higher among males compared to females. More females than males were sedentary (58.7% vs. 48.0%, *p* < 0.001). Moreover, only 12.9% of females performed ≥ 150 min/week of physical activity compared to 24.9% of males (*p* < 0.001). Dietary intake of carbohydrates, proteins, and fat were significantly lower among females compared to males. More than 85% of the study population were on OHA’s with very little sex difference (females: 85.3%; males: 85.9%). Obesity was twice as common in females as compared to males (31.4% vs. 15.6%, *p* < 0.001). Hypothyroidism was noted to be significantly higher in females compared to males (12.5% vs. 3.5%). The frequency of all cancers was higher in females compared to their counterparts (females 1.3% vs. males 0.6%).

Table 2 presents the sex-wise biochemical characteristics of the study population. There were significant differences in glycemic and lipid parameters between females and males. Females had higher FPG, PPPG, HbA1c, serum cholesterol, HDL cholesterol and LDL cholesterol than males. The sex-wise distribution of glycemic parameters stratified based on age group is presented in Figure 1. Among younger individuals, overall, glucose levels were higher than in older individuals, and younger males had higher levels of glucose compared to younger females. However, after the age of 44 years, the control of diabetes was worse among females. Overall, only 18.8% of the females achieved glycemic control (HbA1c < 7%) compared to 19.9% of males (*p* < 0.001).

The most common complication observed in both females and males was neuropathy (females 36.9% vs. males 42.9%, *p* < 0.001), followed by retinopathy (females 26.3% vs. males 36.0%, *p* < 0.001), and nephropathy (females 23.3% vs. males 25.0%, *p* < 0.001). Figure 2a presents the sex-wise frequency of diabetes-related complications stratified based on quartiles of age (years). It was observed that retinopathy, neuropathy, and CVD were higher in males in all age groups. PVD prevalence was generally low in both genders and at >59 years it was higher in males compared to females. The prevalence of nephropathy was significantly higher among males in the ≤46 and >59 years age groups compared to females.

When the prevalence of complications was assessed based on quartiles of duration of diabetes, a similar trend of higher prevalence of retinopathy, neuropathy, and CVD among males was observed (Figure 2b). The prevalence of nephropathy was significantly higher in males with ≥10 years duration of diabetes compared to females, and the prevalence of PVD was higher in males compared to females when the duration was >6 years.

Multiple logistic regression analysis (Table 3) showed that compared to females, males had a higher risk of developing microvascular complications, except nephropathy and macrovascular complications, even after adjusting for confounding factors. Males had a 1.8 times higher risk of developing cardiovascular disease and a 1.6 times higher risk for retinopathy compared to females.

## 5. Discussion

The present study reports the clinical and biochemical profiles, as well as the associated complications of diabetes in a large series of 72,980 age- and gender-matched individuals with T2DM who reported between 2017 and 2019 at a private tertiary diabetes care centre in India. The main findings from this study are as follows: (i) Overall, obesity and hypothyroidism rates were higher in females compared to males; (ii) Physical activity levels were lower among females compared to males; (iii) Control of diabetes was worse among females, especially at older age groups; (iv) Overall, complications of diabetes were lower in females; and (v) Males had a 1.8- and 1.6-times higher risk of developing CAD and retinopathy, respectively, compared to females.

Diabetes is a worldwide epidemic that is rising steadily in both advanced and emerging economies [27]. A successful management outcome for diabetes mellitus is dependent on several factors, including early diagnosis, dietary modification, exercise regimens, maintaining optimal blood glucose levels, and the quick detection and treatment of complications. We found that females had significantly higher HbA1c, FPG, and PPPG levels than males. Earlier studies also reported that females are less likely than males to meet their glycemic targets [28,29]. This can be attributed to the difference in glucose homeostasis, sex hormones, drug response, and psychosocial reasons [30,31]. Poor glycemic control observed in females could also be attributed to the need to balance their treatment with familial responsibilities, which leads to poor treatment adherence as compared to males [32,33].

Obesity and overweight play an important role in the development of T2DM and its complications, which leads to an increased risk of mortality [34,35]. Females with obesity are more likely to develop diabetes, exposing them to a higher risk for cardiovascular disease (CVD) [35]. A recent study from India predicts that the prevalence of overweight and obesity in India is expected to double and triple, respectively, among Indians aged 20–69 years between the years 2010 and 2040 and by 2040, 13.9% of females and 9.5% of males would be affected by obesity in India [36]. Our study also reports a significantly higher prevalence of obesity in females as compared to males (*p* < 0.001).

Efforts for obesity prevention are mostly centered on healthy eating and physical activity [37]. Physical activity aids in weight reduction and subsequently mitigates the risk of having metabolic and orthopaedic conditions [38]. It is well documented that females are less physically active than their male counterparts (31.7% vs. 23.4%) [39]. Females, right from their preschool years, devote less time to physical activity and this persists into their adult years [40]. This not only increases the risk of T2DM but also elevates the risk for cardiovascular disease and cancer, which are the first and second most common cause of mortality in females. Diabetes is currently the seventh most common cause of mortality among females, indicating a link between physical inactivity and disease development [40]. In our study, almost half of the females reported performing no physical activity at all.

Dietary modification plays an important role in the prevention and management of T2DM [41,42]. Evidence suggests that low-carbohydrate diets are effective in the management of diabetes, by significantly reducing body weight and also effectively improving HbA1c, fasting plasma glucose, blood lipid, and insulin resistance [43,44]. In addition, it has been shown that in individuals with diabetes, high-protein diets can significantly decrease HbA1c levels [45], and low-fat intake could improve total cholesterol and LDL cholesterol and lower HDL cholesterol [46]. The ICMR INDIAB national study reported that the total calorie intake was higher in males with newly diagnosed diabetes, while the carbohydrate, protein, and fat intake did not significantly differ by sex [47]. Data from the present study shows that intakes of all macronutrients were lower in females compared to males.

Micro- and macrovascular complications are the major cause of diabetes-related morbidity and mortality [48,49]. These complications pose a great disease burden. Cardiovascular disease is the major macrovascular complication observed with diabetes and is two to three times more common in people with diabetes than those without diabetes [50]. Our study reports that males are at 1.8 times higher risk of developing cardiovascular diseases as compared to females. This is in contrast with various previous studies where an increased frequency of cardiovascular events has been reported in females [51,52,53]. According to some estimates, compared to males, females have a 44% and 27% higher risk of incident coronary heart disease (CHD) and stroke, respectively [54]. Additionally, we report that males were 1.6 times and 1.4 times more prone to develop diabetic retinopathy and diabetic neuropathy, respectively, when compared to their female counterparts in spite of their diabetes control being better. Similar findings have been reported by other researchers as well; however, it is uncertain what causes sex-specific disparities in diabetic microvascular disease [55,56]. It has been hypothesized that differences in the production of inflammatory cytokines along with other factors such as biological dissimilarity in general, lifestyle choices, and treatment adherence could lead to an increased frequency of such complications in males [56].

With the increase in life expectancy in view of advances in diabetes management, there is a shift in complications from conventional ones to unconventional ones like cancer, infections, liver disease, affective disorders, and functional and cognitive disability [57]. Several studies have reported heightened cancer risk in those with diabetes [58]. When compared to the general population, people with diabetes have higher chances of developing cancers of the urinary system, liver and biliary tract, pancreas, colon, endometrium, and kidney [59]. There is also a strong link between obesity and cancer, although this risk varies for different organs [60]. In our study, there was a significantly higher frequency of cancer among females. Future studies should look at the cancer risk in females with T2DM.

We observed a higher frequency of hypothyroidism in females compared to males (12.5% vs. 3.5%). This supports previous studies where a higher prevalence of hypothyroidism was noted among females, and this mandates a more extensive screening for hypothyroidism [61,62,63].

We report that females have poor glycemic control. The period of menopausal transition known as perimenopause, which may begin anywhere from the mid-30s to the mid-50s, may contribute to the worsening of diabetes control [64]. Most females in their mid-life engage in less physical activity and have poor eating habits, which can contribute to weight gain [65]. The menopause transition is associated with a 20–25% weight gain, with the majority of the fat mass concentrated in the abdominal area [66]. During perimenopause, there is an increase in visceral fat, which increases the production of proinflammatory cytokines, circulating free fatty acids, and reactive oxygen species, all of which contribute to insulin resistance [67]. As a result, the menopausal transition is characterised by an increase in total body weight, visceral adiposity, and impairment of glucose homeostasis [67].

In this study, more females received a combination of both oral agents and insulin as compared to males. This finding is in line with an Italian multicentric study, which reported that females were more often managed with combination therapy than their male counterparts [68]. Another study done by Kramer et al. [69] reported that males had less intense pharmacological intervention and contact with their physicians. Other studies have reported contradictory findings and reported that males have more intense pharmacological management compared to females [70,71].

This study highlights female-specific considerations such as the high prevalence of obesity, less physical activity, and concerns about diabetes diagnosis. Interventions that could modify these lifestyle factors can result in better outcomes for T2DM management in women. Considering the trend of poor glucose control during the perimenopausal period, glycemic management must be intensified during this period to prevent or delay diabetes complications in females. The study also points to the need for more frequent screening for cancers in women.

Our study has some limitations. Firstly, being a retrospective study, there was limited information regarding some of the confounding factors (e.g., case-mix and missing data), which could have affected the results. Secondly, as this is a clinic-based study, the findings may not be generalizable to the whole of India, as there could be some referral bias being a private diabetes centre. Despite these limitations, the study has several strengths, including the large sample size and the use of standardized methods to assess diabetes and its complications. The study points to the need for studying gender-related variables in clinical studies. Improving the knowledge regarding diabetes in females is clearly based on the study results.

It would be helpful to examine potential sex variations in various risk factor levels related to glucose metabolism status and across levels of glycemic control in subsequent investigations. This emphasises the critical requirement for sex- and gender-specific risk assessment methodologies and therapeutic interventions that focus on diabetes management in the light of preventing diabetes-related complications. These findings may help to identify specific high-risk groups in clinical settings.

## 6. Conclusions

In conclusion, in this large sample of T2DM, females had higher prevalence of metabolic risk factors and poorer diabetes control compared to males, emphasising the need for better control of diabetes and other metabolic risk factors in females. With regard to complications, males had a higher prevalence of several complications, suggesting that male sex, per se, might be a risk factor for the development of diabetes-associated complications. The study also points to a need for opportunistic screening for hypothyroidism and cancer, especially in females, which can be done at the diabetes clinic to reduce the overall morbidity and mortality among females with diabetes.

## Figures and Tables

**Figure 1 healthcare-11-01634-f001:**
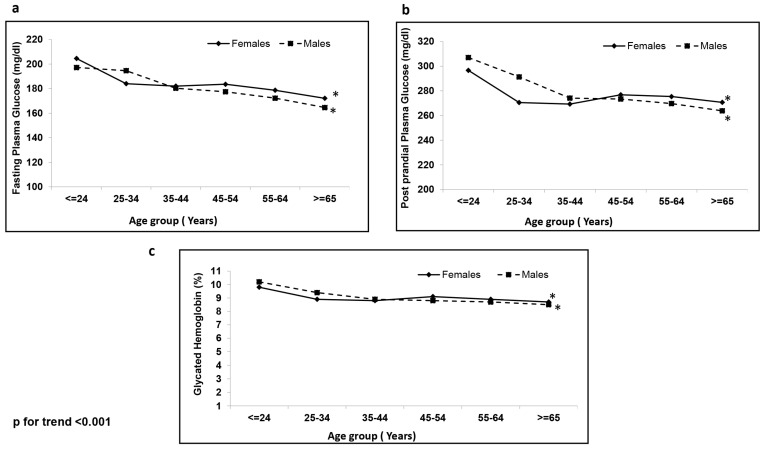
Sex-wise distribution of glycemic parameters stratified based on age group. (**a**) Fasting Plasma Glucose; (**b**) Post Prandial Plasma Glucose; and (**c**) Glycated hemoglobin. *, *p* < 0.01.

**Figure 2 healthcare-11-01634-f002:**
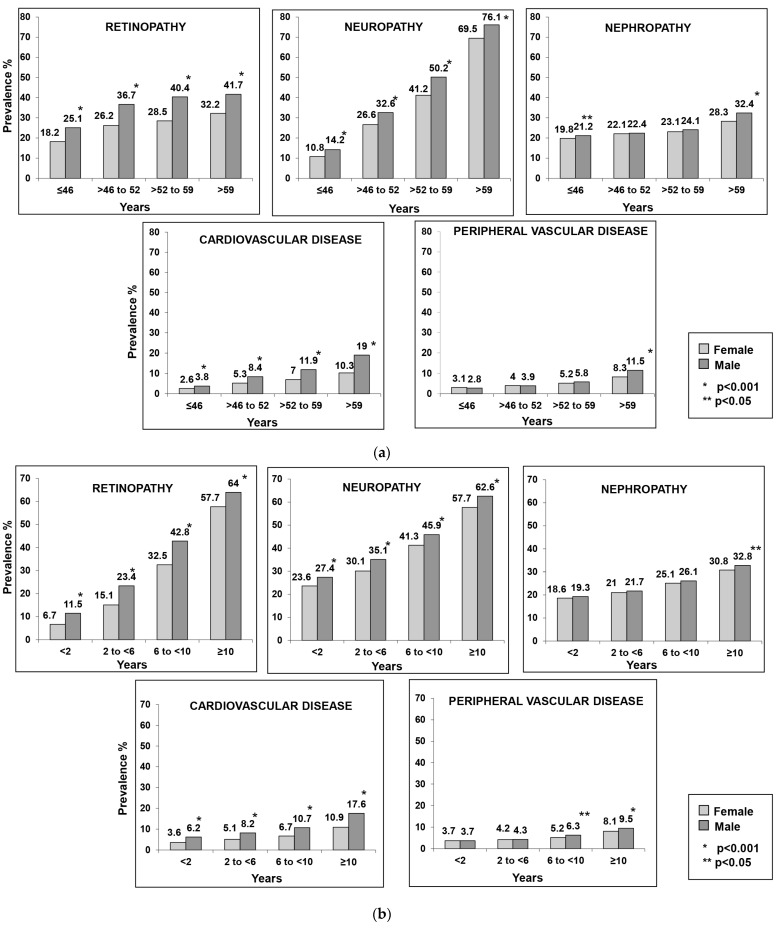
(**a**) Sex-wise frequency of diabetes complications stratified based on quartiles of age (years); (**b**) Sex-wise frequency of diabetes complications stratified based on quartiles of duration of diabetes (years).

**Table 1 healthcare-11-01634-t001:** Sex-wise demographic, clinical, and biochemical characteristics of the study population.

	Males (*n* = 36,490)	Females (*n* = 36,490)
**Age (years)**	53 ± 10	53 ± 10
**Age at onset of diabetes (years)**	45 ± 9	46 ± 10 *
**Height (cms)**	167.2 ± 6.5	153.4 ± 6 *
**Weight (kgs)**	73.5 ± 12.9	66.5 ± 12.4 *
**Body mass index (kg/m^2^)**	26.2 ± 4.1	28.2 ± 4.9 *
**BMI Categories *n* (%)**		
<18.5	400 (1.1)	264 (0.7) *
18.5–24.9	14,534 (39.8)	9393 (25.7) *
25.0–29.9	15,853 (43.4)	15,390 (42.2) *
≥30	5703 (15.6)	11,443 (31.4) *
**Systolic blood pressure (mmHg)**	130 ± 16	130 ± 17
**Diastolic blood pressure (mmHg)**	79 ± 8.4	78.5 ± 8.1
**Duration of diabetes (years)**	7.7 ± 6.8	6.8 ± 6.3 *
**Occupation *n* (%)**		
Home-maker	0	4801 (13.2)
Private employee	14,375 (39.7)	27,601 (76.1) *
Government employee	3019 (8.3)	1654 (4.6) *
Self-employed/Business	13,329 (36.8)	484 (1.3) *
Retired	5417 (14.9)	1662 (4.6) *
Un-employed/Student	105 (0.3)	66 (0.2) **
**Physical activity**		
None	17,518 (48.0)	21,436 (58.7) *
<150 min/week	9892 (27.1)	10,357 (28.4) *
≥150 min/week	9080 (24.9)	4697 (12.9) *
**Smoking *n* (%)**	11,718 (32.1)	348 (1.0) *
**Alcohol *n* (%)**	14,399 (39.5)	64 (0.2) *
**Non-Vegetarians *n* (%)**	28,862 (79.7)	26,115 (72.1) *
**Dietary intake**		
Total calories (Kcals/day)	1367 ± 425	1211 ± 349 *
Carbohydrate (gms /day)	224 ± 72	200 ± 60 *
Protein (gms/ day)	48 ± 14	43 ± 12 *
Fat (gms/day)	32 ± 19	27 ± 16 *
**Management *n* (%)**		
Oral hypoglycemic agents	31,332 (85.9)	31,114 (85.3) **
Insulin	235 (0.6)	189 (0.5) **
Oral hypoglycemic agents + insulin	4923 (13.5)	5187 (14.2) **
**Other comorbidities *n* (%)**		
Cancers	216 (0.6)	476 (1.3) *
Asthma/ chronic obstructive pulmonary disease	180 (0.5)	122 (0.3) **
Hypothyroidism	1292 (3.5)	4549 (12.5) *

* *p* < 0.001; ** *p* < 0.05.

**Table 2 healthcare-11-01634-t002:** Gender-wise biochemical characteristics of the study population.

Biochemical Parameters	Males (*n* = 36,490)	Females (*n* = 36,490)
Fasting plasma glucose (mg/dL)	176 ± 68	180 ± 73 *
Post prandial plasma glucose (mg/dL)	273 ± 94	274 ± 97 **
HbA1c (%)	8.8 ± 2	8.9 ± 2 *
Serum cholesterol (mg/dL)	177 ± 44	187 ± 45 *
Serum triglycerides (mg/dL)	179 ± 133	167 ± 105 *
Serum HDL cholesterol (mg/dL)	39 ± 9	44 ± 10 *
Serum LDL cholesterol (mg/dL)	102 ± 38	109 ± 38 *
Total cholesterol/HDL cholesterol ratio	4.7 ± 1.3	4.4 ± 1.2 *
Blood urea (mg/dL)	25 ± 11	23 ± 10 *
Serum creatinine (mg/dL)	0.9 ± 0.4	0.7 ± 0.6 *

* *p* < 0.001; ** *p* < 0.05.

**Table 3 healthcare-11-01634-t003:** Risk for micro- and macrovascular complications in relation to sex.

Complications	Adjusted Odds Ratio * (95% CI)
**Taking females as reference = 1**	
**Microvascular**	
Retinopathy (yes)	1.59 (1.52–1.66), *p* < 0.001
Nephropathy (yes)	1.03 (0.99–1.08), *p* = 0.169
Neuropathy (yes)	1.35 (1.30–1.40), *p* < 0.001
**Macrovascular**	
Cardiovascular disease(yes)	1.75 (1.64–1.87), *p* < 0.001
Peripheral vascular disease (yes)	1.16 (1.08–1.26), *p* < 0.001

* Adjusted for HbA1c, duration of diabetes, Smoking, Alcohol, Physical activity level, and energy intake.

## Data Availability

The data presented in this study are available on request from the corresponding author. The data are not publicly available due to containing information that could compromise the privacy of research participants.

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
