# Peer review of "Sex-Based Differences in Clinical Profile and Complications among Individuals with Type 2 Diabetes Seen at a Private Tertiary Diabetes Care Centre in India"

_healthcare, 2023, doi:10.3390/healthcare11111634_

Round 1
Reviewer 1 Report (Previous Reviewer 2)
The authors included all the comments and revised the manuscript which can be published with minor English editing.
Author Response
Please see attachment

Reviewer 2 Report (New Reviewer)
I really enjoyed your work, clear, robust and relevant : "Knowing to better manage".
Just 2 points: - The tertiary diabetes care centre in India is a private service, ins't it ? in my country , for example, the public health has 3 levels: primary, secondary and tertiary, so I think it should be clarified from the beginning.
About informed consent, how it was obtained?
Congratulations !
Best regards.
Author Response
Please see attachment

Reviewer 3 Report (New Reviewer)
The work submitted by Pradeepa et al is an interesting observational study that demonstrates sex-based differences in T2DM patients regarding management outcomes in India. The work has interesting implications at the public health level and is thus relevant.
The reviewer would like to offer a few points for the authors to consider fro the improvement of the paper:
1. BMI is an index and as such does not have units. Kg/m2 is a calculation and it does not allude to surface area for example which we do not measure.
2. Please specify the hypothesis of the study in the last paragraph of the introduction section.
3. Please consider specifying the inclusion and exclusion criteria for study participation.
4. How was the number of participant determined (power calculation etc)?
5. What were some confounding factors (smoking, socioeconomic status etc) and how were they normalized?
6. It would be interesting to discuss the management of diabetes through the lens of diet and physical activity. Were there differences in these aspects between men and women in the study described?
Round 2
Reviewer 3 Report (New Reviewer)
The authors have reasonably, addressed the reviewer's comments. Proofreading and care in the formatting is suggested.
Author Response
As suggested by the reviewer, we have now carefully proofread and formatted the manuscript

This manuscript is a resubmission of an earlier submission. The following is a list of the peer review reports and author responses from that submission.
Round 1
Reviewer 1 Report
The study by Pradeepa et al is focused on sex-based differences in clinical profile and complications in patients suffering from type 2 diabetes. The aim of the study is not very clear, and the findings are based on very small differences. The presented data does not support the conclusion. The study design is biased resulting in inaccurate findings: The number of male patients is twice as high as the female, with obvious age differences which results in differences in diabetes complication. The authors should have reduced the male patients to obtain age matched patients in both groups. This approach will avoid all the discrepancy around the age and time of diagnosis.
Results:
Figure 1c and figure 2 (peripheral vascular disease), differences are very small, not sure how they are statistically significant. Student t-test shouldn’t be used for this big study population. Probably due to the huge differences in the number of patients between the two groups.
Discussion:
The discussion is mainly focussed on physical activity and menopause, whereas diabetes complications presented in figure 2 are not discussed sufficiently. Similarly, the type of medication used for diabetes management are not discussed at all. I suggest the authors rethink the whole study design, data analysis and rewrite the paper.
Reviewer 2 Report
What is the cut-off age to get in males and females to get type 2 diabetes?
Some text must be added to discuss the future work or research and how this can be implemented in the clinical setup